# Natural History of *Sudan ebolavirus* to Support Medical Countermeasure Development

**DOI:** 10.3390/vaccines10060963

**Published:** 2022-06-16

**Authors:** Caroline Carbonnelle, Marie Moroso, Delphine Pannetier, Sabine Godard, Stéphane Mély, Damien Thomas, Aurélie Duthey, Ophélie Jourjon, Orianne Lacroix, Béatrice Labrosse, Hervé Raoul, Karen L. Osman, Francisco J. Salguero, Yper Hall, Carol L. Sabourin, Michael J. Merchlinsky, James P. Long, Lindsay A. Parish, Daniel N. Wolfe

**Affiliations:** 1Inserm, Laboratoire P4 Jean Mérieux, 69007 Lyon, France; caroline.carbonnelle@evotec.com (C.C.); marie.moroso@inserm.fr (M.M.); delphine.pannetier@inserm.fr (D.P.); sabine.godard@inserm.fr (S.G.); stephane.mely@inserm.fr (S.M.); damien.thomas@inserm.fr (D.T.); a_duthey@yahoo.fr (A.D.); ophelie.jourjon@inserm.fr (O.J.); orianne.lacroix@inserm.fr (O.L.); beatrice.labrosse@inserm.fr (B.L.); herve.raoul@inserm.fr (H.R.); 2Evotec Id, 69007 Lyon, France; 3UK Health Security Agency, Porton Down, Salisbury, Wiltshire SP4 0JG, UK; karen.osman@ukhsa.uk.gov (K.L.O.); javier.salguero@ukhsa.uk.gov (F.J.S.); yper.hall@ukhsa.uk.gov (Y.H.); 4U.S. Department of Health and Human Services (DHHS), Tunnell Government Services, Inc., Supporting Biomedical Advanced Research & Development Authority (BARDA), Assistant Secretary for Preparedness and Response (ASPR), Washington, DC 20201, USA; carol.sabourin@hhs.gov (C.L.S.); james.long@hhs.gov (J.P.L.); 5U.S. Department of Health and Human Services (DHHS), CBRN Vaccines, Biomedical Advanced Research & Development Authority (BARDA), Assistant Secretary for Preparedness and Response (ASPR), Washington, DC 20201, USA; michael.merchlinsky@hhs.gov (M.J.M.); lindsay.parish@hhs.gov (L.A.P.)

**Keywords:** *Sudan ebolavirus*, SUDV, Ebola, natural history study, filovirus, macaques, telemetry, virus

## Abstract

*Sudan ebolavirus* (SUDV) is one of four members of the *Ebolavirus* genus known to cause Ebola Virus Disease (EVD) in humans, which is characterized by hemorrhagic fever and a high case fatality rate. While licensed therapeutics and vaccines are available in limited number to treat infections of Zaire ebolavirus, there are currently no effective licensed vaccines or therapeutics for SUDV. A well-characterized animal model of this disease is needed for the further development and testing of vaccines and therapeutics. In this study, twelve cynomolgus macaques (*Macaca fascicularis*) were challenged intramuscularly with 1000 PFUs of SUDV and were followed under continuous telemetric surveillance. Clinical observations, body weights, temperature, viremia, hematology, clinical chemistry, and coagulation were analyzed at timepoints throughout the study. Death from SUDV disease occurred between five and ten days after challenge at the point that each animal met the criteria for euthanasia. All animals were observed to exhibit clinical signs and lesions similar to those observed in human cases which included: viremia, fever, dehydration, reduced physical activity, macular skin rash, systemic inflammation, coagulopathy, lymphoid depletion, renal tubular necrosis, hepatocellular degeneration and necrosis. The results from this study will facilitate the future preclinical development and evaluation of vaccines and therapeutics for SUDV.

## 1. Introduction

There are six known virus species within the genus Ebolavirus including Bundibugyo virus, Sudan virus (SUDV), Tai Forest virus, and Zaire ebolavirus (EBOV), known to infect and cause Ebola virus disease (EVD) in humans [1,2]. EVD is characterized in humans by hemorrhagic fever and other general symptoms including fever, headache, malaise, diarrhea, and vomiting [3]. The case fatality rate of EVD can vary between species of Ebolavirus, SUDV having an average of 53% mortality in documented outbreaks [4]. The majority of the EVD outbreaks in humans to date have been caused by EBOV with SUDV being the second leading cause of EVD [4,5]. The first recorded outbreak of SUDV occurred in South Sudan in June 1976 [6]. Since then, there have been six sporadic outbreaks of SUDV, all occurring in South Sudan and Uganda, with the last known outbreak occurring in 2012 [4].

The severity and rapid spread of the 2014–2016 West Africa Zaire EBOV epidemic highlighted the necessity of countermeasures to protect against filoviruses of high consequence. While there are a few licensed vaccines available for the prevention of EBOV EVD [7,8], there remains an urgent unmet need for safe and efficacious vaccines and therapeutics against SUDV. However, the sporadic nature of SUDV outbreaks presents a challenge to evaluating the efficacy of vaccine and therapeutic candidates in human clinical trials, which is the common regulatory path for licensure and approval by the Food and Drug Administration (FDA). To address these challenges, the FDA has developed the Animal Rule (See 21 CFR 314.600 for drugs and 21CFR 601.90 for biological products) to allow for nonclinical efficacy data to substitute for human clinical data when human clinical studies are not feasible. To achieve vaccine or therapeutic licensure through the animal rule, the FDA guidance requires that, “The effect is demonstrated in more than one animal species expected to react with a response predictive for humans, unless the effect is demonstrated in a single animal species that represents a sufficiently well-characterized animal model for predicting the response in humans” [9,10]. In particular, cynomolgus macaques (*Macaca fascicularis*) have been the most relevant animal model for evaluation of candidate vaccines against EVD and the only one that has been demonstrated to predict efficacy in humans. Indeed, vaccine efficacy against EVD caused by EBOV in this model correlated closely with clinical efficacy observed in humans, both in terms of levels of protection and onset to protection ([11,12], reviewed in [13]). Thus, we decided to investigate if a similar nonhuman primate model was a suitable predictive model for the evaluation of SUDV medical countermeasures. Previous studies challenging nonhuman primates with SUDV suggested that a nonhuman primate model would be an appropriate model. However, these studies either did not fully characterize the pathogenesis of the disease, record telemetry data in untreated animals, or they evaluated routes other than IM for challenge [14,15,16,17,18,19,20,21]. This natural history study was designed to investigate feasibility of nonhuman primates as model for the Animal Rule using a sufficient number of animals (*n* = 12) to provide statistical power and utilized comprehensive monitoring to investigate the course and reproducibility of the infection in these animals. For the challenge strain, we chose to use SUDV Gulu as this isolate satisfies key requirements for use as challenge material to evaluate medical countermeasures by the Animal Rule [22].

The progression of SUDV disease in cynomolgus macaques was documented by monitoring and recording the clinical signs of disease after intramuscular (IM) injection of 1000 plaque forming units (PFUs) of SUDV in twelve healthy, research-naïve cynomolgus macaques. Many biomarkers characteristic of SUDV-challenged cynomolgus macaques were monitored and recorded, including mortality rate and time to euthanasia, time from exposure to onset of disease clinical signs (such as weight loss), clinical pathology readings, time course and progression of disease characteristics, and frequency and magnitude of disease signs. All animals exhibited viremia and clinical signs consistent with SUDV infection by Day 4 post-challenge. These clinical signs included fever, reduced physical activity, gastrointestinal signs like diarrhea, dehydration, and macular skin rash. At post-mortem examination, lesions included systemic inflammation, coagulopathy, lymphoid depletion, renal tubular necrosis, hepatocellular degeneration and necrosis. Overall, this natural history study demonstrates that SUDV infection in cynomolgus macaques results in systemic viremia sharing many characteristics of SUDV infection in humans, thereby establishing the cynomolgus macaques as a SUDV challenge model for the evaluation of vaccines and therapeutics.

## 2. Materials and Methods

### 2.1. Ethics Statement

Animal research was conducted at the P4 Inserm BSL4 animal facility, which is accredited for experimentation using nonhuman primates by the French authorities, “Direction départementale des services vétérinaires” (Ministry of Agriculture). The study was implemented with approval from the ethical committee CELYNE N°42 (Comité d’Ethique Lyonnais pour les Neurosciences Expérimentales) and the Ministry of Research (APAFIS; authorization No.APAFIS#26121-2020061914462766v4). P4 Inserm Animal Welfare Assurance has been approved by the US Office of Laboratory Animal Welfare (OLAW), identification number F20-00468.

### 2.2. Animals

Animals purchased for this study came from an Association for Assessment and Accreditation of Laboratory Animal Care International (AAALAC International) accredited source. Six male and six female cynomolgus macaques (*Macaca fascicularis*, Asiatic origin bred in Vietnam) weighing between 2.36–2.83 kg and between 2.43–2.74 years old at the time of challenge were used for this study. Health records indicated that the animals were negative for tuberculosis, herpes B virus, retroviruses, SIV (Simian immunodeficiency virus), STLV (Simian T-lymphotropic virus), SRV-D (Syncytial Respiratory Virus Disease) and pathogenic enterobacteria (Shigella, Yersinia, Salmonella). Animals selected for inclusion in the study were confirmed negative for pre-existing anti-filovirus glycoprotein antibodies by ELISA (Recombvirus TM Monkey Anti-Ebola Virus Glycoprotein Combo IgG ELISA Kit, Catalog Number AE-325600-XM Gentaur Ltd., used as per manufacturer’s instructions). Each animal was identified by an original tattoo (unique nine-digit identification number).

Animals were handled according to the European regulations (European Directive 2010 63/UE) and the strict procedures imposed for work in high security BSL-4 containment. All work involving animals was performed according to standard operating procedures established for animal experiments in BSL-4 facilities. The facility has central air conditioning equipment which maintains a constant temperature of 22 ± 2 °C. Air is renewed at least 20 times per hour in animal rooms. Animals were kept in isolators specifically designed for BSL4 areas, which strictly comply with European regulations in terms of floor surface per animal. The isolators were maintained under negative pressure and absolute air filtration. The animals were housed in groups of three (grouped by gender) in four cabinets that allowed social interactions. Before challenge, the animals were acclimatized for twelve days following a seven-day baseline period in BSL4 conditions with enriched sterile environment, access to food and water, and light/dark cycles. Baseline telemetry data were collected from the acclimation period. Cages were placed in ventilated safety cabinets equipped with HEPA filters, in depression under 40 Pa. The temperature was targeted to be maintained between 20 °C and 28 °C with a relative humidity of 40% to 70%. Animals were maintained on 12 h light/12 h dark photoperiod, except when room lights were turned on for study-related procedures (animal observations). Each cabinet included an elevated perch or platform, and each animal was provided with toys for physical enrichment and was provided with audio stimulation. Animals were fed with a commercial pellet diet from SDS Dietex (OWN banana 808003) in three bowls of 130 g each day per cage. In addition to pellets, the animals were provided with two fresh fruits per animal per day. Tap water was provided as three bottles of 730 mL each day per cage.

### 2.3. Surgical Procedures 

To acquire body temperature and activity data, telemetry DSI M00 (Data Sciences International (DSI), Silabe, Strasburg) as well as StarOddi micro-T transmitters were surgically implanted in the peritoneum by a veterinary surgeon into each animal 55 days (for 10 out of 12 animals) and 54 days (for the two remaining animals) prior to SUDV challenge. The health status of the animals was monitored by a veterinary surgeon.

### 2.4. Randomization

There was no randomization or blinding. All twelve animals were challenged with the same volume from the dilution of the SUDV stock dilution. No unchallenged controls were used in this study.

### 2.5. Body Temperature

Rectal temperature monitoring was performed at each day of sampling on sedated animals on Day −7, Day −4, Day 0 (before challenge) and Day 3, Day 5, Day 7, and Day 10 as well as before each euthanasia.

### 2.6. Telemetry Data Collection and Analysis

#### 2.6.1. DSI PhysioTel Digital M00 Series

PhysioTel digital M00 telemetry devices (DSI) were surgically implanted 54 or 55 days prior to exposure. Surgical implantation of the devices was performed by the NHP (non-human primates) supplier (Silabe, Strasburg). The units transmitted temperature and activity beginning at the day of arrival of the animals until the day of euthanasia of the animal.

During acquisition, data was transmitted from the implant to a TRX-1 receiver mounted in the room connected via a Communications Link Controller (CLC, DSI) to allow the digital multiplexing and the simultaneous collection of signals from all animal subjects. Digital data were then captured, reduced, and stored using the Ponemah Acquisition Software (version 5.32, DSI). Reduced data were extracted and all time points for all animals have been considered for the analysis of each parameter for each animal.

Activity measurement was reported by the Ponemah software, which used a three-axis accelerometer contained in the PhysioTel Digital implants. The three-axis accelerometer provides acceleration data along the x-, y-, and z-axes, relative to the orientation of the implant. Acceleration for the x-, y-, and z-axes is reported as a value from an analog to digital converter. Along with the values from each axis of the accelerometer, Ponemah software also reports an Activity value calculated from the accelerometer axes in Jerks.

At the end of the study, all DSI real-time activity data were exported and analyzed taking into account the average activity over the defined logging rate, normalized to a minute and expressed as A_MPMN (Activity Non Pulsative Mean).

DSI temperature was read daily for each animal. At the end of the study, data was exported for analysis. All pre-challenge temperature data from the BSL4 acclimation period starting at Day −7 was used to develop a baseline. Telemetry real-time temperature data collected prior to this period was used as baseline and provided the baseline minimum and baseline maximum average body temperature. All points of post-challenge data were analyzed and reported from the baseline to the end of the study. Data were assessed for quality and were omitted if the signals were not of sufficient quality to be analyzed. Data were omitted in the cases of signal drop out and/or non-physiological values.

Derived parameters were compared to the baseline average. Fever onset, fever duration, fever hours (sum of significant temperature elevations), and average fever elevation (fever hours divided by fever duration in hours) were computed. Parameters evaluated were temperature, temperature mean, activity, and activity mean.

#### 2.6.2. Star Oddi DST Micro-T Transmitters

Star Oddi DST micro-T transmitters were implanted 55 or 54 days prior to exposure and pre-programmed to record temperature every ten minutes against UK BST (UTC +1). Surgical implantation was performed by the NHP supplier (Silabe, Strasbourg). At the end of the study, the implants were retrieved, and the temperature data uploaded to tabular format using the Star-Oddi Mercury Data Logger PC Software, according to the manufacturer’s instructions. To allow a qualitative comparison between DSI- and StarOddi-derived data, an identical analysis was performed. Therefore, all pre-challenge temperature data from the BSL4 acclimation period starting at Day −7 pre-challenge were used to develop a baseline period. Post-challenge data were analyzed and were reported from baseline to the end of the recording period for all animals. Derived parameters were compared to the baseline average. Fever onset, fever duration, fever hours, and average fever elevation was determined.

### 2.7. Challenge Material

The challenge stock used in this study was a dilution of *Sudan ebolavirus* (SUDV), Gulu, H. sapienstc/2000 working stock (BEI stock NR50733) sourced from the Biodefense and Emerging Infections Research Repository, USA. The challenge stock was a third cell culture passage (P3) and had a 100% 7-uracil (U) genotype at the glycoprotein editing site start at nucleotide 6925. Each 1 mL contained 2.5 × 10^6^ PFU *Sudan ebolavirus* Gulu 2000 in clarified infected Vero E6 supernatant plus 10% fetal bovine serum. On the day of challenge, the virus stock was removed from frozen storage and allowed to thaw under ambient conditions. The thawed suspension was diluted in PBS (Cat. No. 14190-169, Gibco Life Technologies, Carlsbad, CA, USA) in two steps: a first 1:10 dilution followed by a 1:125 dilution, such that animals received a target dose of 1000 PFUs per 0.5 mL. Confirmation of infection with live virus was obtained by parallel back titration of the viral stock and challenge inoculum on the day of challenge, using a focus-forming assay. Samples were from the SUDV challenge dilution prior to inoculation of the first cynomolgus macaque and after challenge of the last cynomolgus macaque. 

### 2.8. Challenge

All animals were anesthetized at Day 0 via intramuscular (IM) injection using tiletamine/zolazepam Zoletil^®^ and then inoculated with a dilution corresponding to 1000 PFUs of SUDV diluted in PBS in a final volume of 0.5 mL. Skin area covering the right *quadriceps femoris* were shaved, and the inoculation site was identified using a permanent marker to facilitate identification post-challenge.

### 2.9. Blood Collection and Processing

Blood collections were performed under sedation via the femoral route using a Vacutainer^®^ system (P-A4-25). Blood samples were collected on Day −7, Day −4, and Day 0 before challenge and on Day 3, Day 5, Day 7, and Day 10 post-challenge. Animals euthanized at an earlier timepoint underwent blood collection prior to necropsy.

Blood which was required to be added to EDTA tubes was added as soon as practically possible and mixed by repeated inversion before further processing. EDTA tubes (BD vacutainer Tubes Cat. No. BD368841, Ozyme) were used at 2 mL total maximum volume at each time of sampling for hematological parameters and plasma samples. Dry Tubes (Cat. No. LABELIANS VK052SAS) were used at 2 mL total maximum volume at each time of sampling for biochemical parameters. S-Monovette tubes (Cat. No. STARSTEDT 57229001) were used for coagulation parameters.

### 2.10. Bioanalytical Testing

All handling and storage of infectious material was carried out in the BSL4 laboratory. Infectious and molecular viral loads in plasma were determined for each day samples were taken as well as the days euthanasia and necropsy were performed.

### 2.11. Determination of Infectious Viral Load by Plaque Assay

Confirmation and quantification of live virus in the Sudan viral stock and dilution used for challenge was accomplished by titration on Vero E6 cells via an immune-detection plaque assay. Infectious viral load was determined using twelve-well microplates of Vero E6 cells. Briefly, the cells were incubated with tenfold serial dilutions of each sample for 1 h at 37 °C, then incubated in the presence of carboxy-methyl-cellulose (CMC) for seven days at 37 °C and 5% CO_2._ After this incubation, cells were first fixed with 3.7% formaldehyde solution and permeabilized with 0.5X Triton. Mouse anti-Ebola GP specific antibody was then incubated 1 h at 37 °C. Second alkaline phosphatase-conjugated polyclonal anti-mouse IgG (A3562-1ML, Sigma Aldrich, St. Louis, MO, USA) was then incubated 1 h at 37 °C before colorimetric revelation of viral foci using 1-step NTB/BCIP plus suppressor (34070, Thermo Scientific, Waltham, MA, USA). Colored viral foci from infectious particles of virus were then counted to quantify the number of infectious viral particles initially present in the sample and the infectious viral titer was expressed as focus-forming units (FFU) per mL of sample.

### 2.12. Determination of Molecular Viral Load by qRT-PCR

Quantitative real-time polymerase chain reaction (qRT-PCR) was performed outside the BSL4 laboratory on inactivated samples. Briefly, 100 µL of sample was mixed with 400 µL of AVL buffer viral lysis (ref 190373, QIAGEN). After ten minutes at room temperature 400 µL of ethanol was added. Viral RNA was then extracted using the QIAamp Viral RNA Mini Kit (Cat. No./ID: 52906) following the manufacturer specifications (QIAGEN). The qRT-PCR reaction was performed on eluates containing viral nucleic acids using specific primers and probes overlapping with the SUDV NP gene and using the BioRad CFX 96 Touch Real Time PCR System (BioRad Laboratories, Hercules, CA, USA; P/N 172-500).

### 2.13. Next Generation Sequencing

Viral RNA from the NHP clinical samples (plasma) taken at Day 5 post-challenge and on the day of necropsy was converted to cDNA using random hexamer primers and the superscript IV reverse transcriptase (Invitrogen by Life Technologies, Carlsbad, CA, USA) using a MasterCycler thermal system (Eppendorf, Hambourg, Germany). Primal Scheme tool was used to design 109 primer pairs covering the whole genome of SUDV strain Gulu 2000 (GenBank Sequence Accession: MH121163). Overlapping 250 bp amplicons were then generated by multiplex PCR using NEB Ultra II Q5 Master mix (New England Biolabs, Ipswich, MA, USA). The run was performed on CFX96 Touch (Biorad, Hercules, CA, USA). After PCR reaction, amplicons were cleansed using Agencourt AMPure XP beads (Beckman Coulter, Brea, CA, USA). Nextera XT DNA Library Preparation and Nextera XT Index Kits (Illumina, San Diego, CA, USA) were used for DNA-Sequencing library construction. Template amplicons were quantified on Qubit 2.0 (Thermo Scientific, Waltham, MA) and normalized at 0.2 ng/µL. Adapters sequences were tagged to the amplicons during the tagmentation process. Indexes were added to the tagmented DNA and a PCR was performed on CFX96 Touch. Libraries were purified using Agencourt AMPure XP beads. The size of the fragments was checked on the Tapestation 42000 device (Agilent technologies, Santa Clara, CA, USA) and the concentration of the library was determined. Libraries were pooled together at equal quantity before being denatured and diluted at the loading concentration (1.4 pM). Sequencing was performed on a Miniseq sequencer (Ilumina, San Diego, CA, USA).

### 2.14. Bioinformatics Analysis

Original FASTQ data was mapped to GenBank reference MH121163 using BWA MEM v0.1.17 with default parameters [23]. iVar v1.3 [24] was used to trim primer sequences and Samtools v1.11 [25] was used to convert, sort, and index trimmed bam files. Consensus sequences were produced from the trimmed bam files using QuasiBam, an in-house C++ program that records base frequencies at each position of a reference. Single nucleotide polymorphisms (SNPs) were called into consensus when contained in 80% of the reads, ambiguous base pairs were called where 20% of reads contained a different base to the reference with a minimum read depth of 100. Consensus sequences were aligned using clustal omega [26]. Further investigation into levels of minority variants contained within the sample was carried out by manual manipulation of the QuasiBam variant data.

### 2.15. Clinical Pathology

All work and storage of blood was performed in the BSL4 laboratory.

#### 2.15.1. Biochemical Analyses

Dry tubes were used to sample blood and centrifuged at 3000 RPM for ten minutes at room temperature. Serum was then collected for the following biochemical analyses: enzymes (ALP, ALT, AST, CK NAC), substrates (total bilirubin, creatinine, and urea) and CRP using the Pentra C200 device (HORIBA Medical PENTRA C200). Controls and calibration reagents were used (HORIBA Medical ALP A11A01626, ALT A11A01627, AST A11A01629, CK NAC A11A01632, TBil A11A01639, Create A11A01933, Urea A11A01641, CRP A11A01611). The remaining volume of serum was aliquoted and stored at ≤−60 °C.

#### 2.15.2. Hematology

Full hematology analysis was performed directly on whole blood stored in EDTA tubes, using a MS9-5 unit (Melet Schloesing Laboratories 3MS09007) with a MS9 pack (MSLabo 3MSR0905) and a control sample tube (MSLabo OMSLC011). The parameters recorded were hematocrit, hemoglobin, red blood cells, thrombocytes, white blood cells and their subsets. After analyses, plasma was separated from remaining whole blood, aliquoted and stored at ≤−60 °C.

#### 2.15.3. Coagulation

Total blood was used to determine blood clotting time (PT for Prothrombin Time, IND for International Normalized Ratio and ACT for activated Clotting Time) using an I-Stat analyzer unit (ABBOTT- 4p75-01). Analyses were certified by control tests performed every week (PT.INR ABBOTT -3P89-24, PT control level 1 ABBOTT -6P17-13, PT control level 2 ABBOTT-6P17-14, ACT ABBOTT-3P86-25, ACT control level 1 ABBOTT-6P17-15, ACT control level 2 ABBOTT 6P17-16).

### 2.16. Clinical Observations

Clinical observations were documented on the appropriate forms by the P4 Inserm animal experimentation team, trained to document individual animal behaviors and appearance. Animals were observed once daily from Day −19 to Day 4 post-challenge. At the first observation on Day 4 post-challenge, one animal (966) was assigned a score of eight, initiating an automatic increase in the frequency of observations to three per day (8 h ± 3 h) for all animals. The late time point involved monitoring via CCTV combined with real-time telemetry readouts for temperature and activity. This frequency of observation was maintained until all animals were humanely euthanized. 

Animals were evaluated cage-side for signs of illness including but not limited to global behavior, condition of stool, physical aspect, dehydration, epistaxis, interaction, tonus and bleeding prior to observations under anesthesia. Weight and rectal temperature monitoring were performed at each anesthesia (Day −7, Day −4, Day 0, Day 3, Day 5, Day 7 and Day 10) and at each time of euthanasia. Physical observations of animals under anesthesia occurred on Day −7, Day −4, Day 0 (before challenge), Day 3, Day 5, Day 7, Day 10 and when each animal was euthanized. 

Animals were monitored at least once per day after challenge. Scores were calculated each day in the morning from day 0 to 3, and from then on calculated in the morning, afternoon and night when data were available. Weight and clinical observations were recorded on the days which animals were sedated. A clinical disease score ranging from 0 to 15 was assigned after summing the scores from observation of weight loss (expressed as a change from Day 0), DSI body temperature (increase or decrease relative to Day 0), behavior, activity, facial expressions of pain, difficult gait, the presence of petechiae and bleeding (Table 1). For the one animal with a non-functional DSI implant, rectal temperature was used. On days where sedation was not performed, weight and observations recorded during the last sedation were carried over to the current time point.

### 2.17. Euthanasia

The pre-determined conditions that warranted euthanasia were a sum of all clinical values that equaled or exceeded 15, a demonstration of persistent prostration, body temperature below 35.8 °C, or an absence of awakening 150 min after anesthesia. Euthanasia was performed by intracardiac administration of sodium pentobarbital under deep anesthesia.

### 2.18. Pathology

#### 2.18.1. Necropsy & Gross Pathology

All animals underwent a complete post-mortem examination following a standard operation procedure. An individual animal necropsy record was completed for each animal. Prior to necropsy, animals were exsanguinated under terminal anesthesia to provide blood samples and then euthanized prior to collection of tissues. During necropsy, gross pathology evaluation was performed to evaluate lesions compatible with hemorrhagic fever. 

#### 2.18.2. Tissue Collection, Organ Weights, and Preservation

Sections of the heart, liver, lungs, kidney, spleen, and inguinal lymph nodes, no greater than 0.5 cm thick, were collected and fixed by immersion in 10% neutral buffered formalin (NBF). Fixed samples were transferred to fresh NBF after a minimum of seven days. Sections of each organ were also individually collected in two cryotubes and stored at ≤−60 °C. Pots containing fixed samples in NBF were decontaminated prior to removal from the BSL4 and then transferred to the UKHSA Pathology laboratory.

#### 2.18.3. Histopathology

Following fixation, samples were processed into paraffin wax and sectioned for hematoxylin and eosin staining. An in situ hybridization method was also applied to detect SUDV viral RNA. Briefly, tissues were pre-treated with hydrogen peroxide for ten minutes at ambient temperature, antigen retrieval for 15 min (95 °C) and protease III for 15 min (40 °C) (Advanced Cell Diagnostics, Newark, CA, USA). A Sudan virus NP-S probe targeting the NP549-2033 gene region (Advanced Cell Diagnostics) was incubated on the tissues for 2 h at 42 °C. Amplification of the signal was carried out following the RNAscope protocol (ACD 2.5 Red Rev B—adapted to use the Sudan virus NP-S probe) using the RNAscope 2.5 LS reagent kit-red (Advanced Cell Diagnostics) and the BOND Polymer refine RED detection system for chromogenic visualization and hematoxylin counterstain (Leica Biosystems, Wetzlar, Germany). Sections were digitally scanned and reviewed by a veterinary pathologist.

### 2.19. Quality System

As intended, this study was not conducted in compliance with FDA GLP 21CFR. The study was conducted in accordance with the respective Quality Management Systems of UKHSA and P4 Inserm that are compliant with BS EN ISO 9001:2015.

Work was conducted in accordance with the UKHSA Quality Assurance Project Plan, Study Plan, associated amendments, applicable SOPs, as well as good documentation and error correction practices.

Internal quality control checking was performed on raw data and worksheets during and after completion of the study. All steps involving transcription of the data were checked by an operator not involved with the transcription process and documented by a dated signature with a statement to the effect that the data are accurate. This evidence of data checking was scanned and stored in the electronic study folder and the originals kept in the study file.

### 2.20. Statistical Analysis

All twelve animals in this study were challenged with SUDV. Thus, there was no statistical comparison. Survival time was expressed as hours post-infection. A Kaplan–Meier plot was generated to determine the median time to euthanasia. Weight loss was expressed as a percentage relative to weight on the day of challenge. Baseline temperature for each NHP was calculated according to the average of all temperature recordings taken prior to challenge. Fever and hyperpyrexia were defined as periods of increase from the baseline of ≥1.7 °C and ≥3 °C, respectively. Baseline, baseline minimum and baseline maximum average body temperatures (BT) were calculated from the BSL4 acclimation period starting at Day −7 pre-challenge. Baseline hematology, biochemistry and coagulation levels were calculated according to the average values taken from all animals pre-challenge.

## 3. Results

### 3.1. Mortality

For this study, twelve cynomolgus macaques were challenged via intramuscular (IM) inoculation with 0.5 mL from a dilution of the starting SUDV stock corresponding to 1000 PFU. Confirmation of infection was obtained through back titration. All twelve SUDV infected animals reached pre-determined clinical scores warranting euthanasia during the study with intervention occurring as early as Day 5 and up to Day 10 post-challenge. Animals were euthanized on Days 5 (*n* = 1), 7 (*n* = 5), 8 (*n* = 1), 9 (*n* = 2) and 10 post-challenge (*n* = 3). Survival times, i.e., the time post-challenge that the clinical scores reached the level commensurate with euthanasia criteria, are summarized below in Table 2. The cause of death for all animals in the study was attributed to SUDV infection.

The median survival time was 182 h post-challenge. A Kaplan–Meier plot is shown in Figure 1. The median time as estimated by the product limit algorithm for right censored data was 188.83 h (7.86 days post-challenge) with an interquartile range of 165.67 to 236 h (6.9–9.8 days post-challenge).

### 3.2. Clinical Scores, Body Temperature, and Weight

#### 3.2.1. Clinical Scores

After infection, all animals were observed at approximately the same time each day and clinical scores were recorded. On the days the animals were sedated, additional clinical signs including weight changes and rectal temperature were included in the observations (Day−7, Day −4, Day 0 (before challenge) Day 3, Day 5, Day 10, and at each time of euthanasia). All clinical scores are shown in Figure 2 and the clinical scoring parameters are outlined in Table 1. Signs of disease included fever, reduced responsiveness, petechial rash, dehydration, bleeding and weight loss. Overall, the severity of the clinical signs in animals increased over time.

The first signs of the disease were observed in one animal (882) on Day 3 with the onset of fever. The remaining animals had an onset of sustained fever starting on Day 4. Additionally, starting on Day 5 post-challenge, two out of twelve animals displayed hunched posture. On Day 3 post-infection diarrhea was observed. Petechial rash was observed in one animal on Day 5 and on Day 7 for seven animals. Dehydration was observed in ten animals at Day 5. Epistaxis was observed on Day 7 in one animal and in a second animal on Day 9. All twelve animals reached the predetermined clinical score warranting humane euthanasia (score ≥ 15) between Day 5 and Day 10 post-challenge. A summary of data for clinical scoring is presented in Appendix A.

#### 3.2.2. Body Temperature

To monitor temperature and activity, all NHPs were implanted with two telemetry systems, the PhysioTel digital M00 (DSI) and the StarOddi DST micro-T transmitters. At the time of activation, excellent signal quality was observed for the majority of the DSI implants. However, one implant failed prior to challenge and did not allow for real time monitoring of internal temperature and activity (animal 778) throughout the experiment. The rectal temperature was used to determine onset of fever in this animal. Temperature data for each of the other eleven animals were continuously collected from the acclimation period and the data collected for at least seven days prior to challenge were used at the end of the experiment to establish a baseline value to compare to real time data for post-challenge temperature monitoring and onset of fever. DSI data were recorded daily.

Rectal temperature was recorded when each animal was sedated. For the one animal for which the DSI implant was non-functional, rectal temperature was used for the assignment of a clinical score. In the intervals between sedation, the rectal temperature recorded during the last sedation was considered the current temperature. The average onset of significant rectal temperature change was 120 h post-challenge (Figure 3).

Baseline, baseline minimum, and baseline maximum average body temperatures were calculated from the ABSL4 acclimation period starting at Day −7 pre-challenge. Post-challenge data were analyzed and were reported from baseline to the end of the recording period for all animals. Derived parameters were compared to the baseline average. Onset of fever post-challenge was defined as the time at which the temperature increased greater than 1.7 °C above the baseline average value.

DSI body temperature data analysis revealed that all animals had a sustained increase of temperature (Figure 4). Significant increase in body temperature from the baseline started from 61.85 h post-challenge for one animal (≥1.7 °C compared to the baseline average values). All SUDV-challenged animals carrying functioning DSI implants developed sustained fever on or before Day 3 post-challenge (range 61.85–96.04 h post-challenge). The onset of fever was first captured in the clinical scoring at the observation point ending at 70.58 h post-challenge.

Sustained hyperpyrexia (≥3 °C compared to the baseline for longer than 2 h) occurred in all animals starting at the range of 91.00–145.20 h after challenge, with four animals developing hyperpyrexia from Day 3 post-challenge, five animals developing hyperpyrexia from Day 4 post-challenge, one animal developing hyperpyrexia from Day 5 post-challenge and one animal developing hyperpyrexia from Day 6 post-challenge.

#### 3.2.3. Temperature Assessment by Star Oddi DST Micro-T Transmitters

At the end of the study, all StarOddi data were exported and analyzed. Baseline, baseline minimum and baseline maximum average body temperatures were calculated from the ABSL4 acclimation period starting at Day −7 pre-challenge as was done for DSI telemetry. StarOddi body temperature analysis revealed that all animals showed sustained increase of temperature. Significant increase in body temperature from baseline started from 61.83 h post-challenge for one animal (≥1.7 °C compared to the baseline average values). All SUDV-challenged animals developed sustained fever on or before Day 3 post-challenge (range 61.83–95.17 h post-challenge).

Sustained hyperpyrexia (≥3 °C compared to the baseline for longer than 2 h) occurred in all animals starting at the range of 90.50–148 h after challenge with two animals developing hyperpyrexia from Day 3 post-challenge, six animals developing hyperpyrexia from Day 4 post-challenge, two animals developing hyperpyrexia from Day 5 post-challenge and two animals developing hyperpyrexia from Day 6 post-challenge (Figure 5).

#### 3.2.4. Weight

Most of the animals exhibited a gradual progressive weight loss (Figure 6). In five out of twelve animals, weight loss of more than 10% compared to Day 0 was observed before euthanasia criteria were met. Additionally, a 7.5 to 10% weight loss was observed in another five animals and only two animals did not show any statistically significant weight changes as compared to their baseline (≤5% weight loss compared to Day 0).

### 3.3. Clinical Pathology

#### 3.3.1. Clinical Chemistry

The effects of SUDV infection on clinical pathology were investigated using serum from blood collected from sedated animals and processing it in a Pentra C2000 device. Creatinine and urea levels were measured to assess renal function in all animals. Both creatinine and urea concentration increased late in infection starting around 120 h and 168 h post-challenge, respectively. Average creatinine and urea concentrations peaked around 216 h post-challenge (Figure 7 and Appendix A).

All animals demonstrated moderate to marked elevations of these parameters, with a mean 3.5 fold change from the baseline at the time of euthanasia for urea concentration and a mean of 4.1 fold change from the baseline at the time of euthanasia for creatinine concentration. In particular, two animals (837 and 986) showed significant increases on the day of euthanasia (fold change from baseline of the urea concentration at the time of euthanasia of 10.1 and 8.8, respectively, and fold change from baseline of the creatinine concentration at time of euthanasia of 15.3 and 7.7, respectively), suggesting that these animals developed a renal functional impairment and dehydration.

Alanine transaminase (ALT), aspartate transaminase (AST), alkaline phosphatase (ALP) and total bilirubin were measured throughout the disease course to assess the degree of liver damage (Figure 7). Generally, the level of these enzymes increased in animals during infection with significant increases in ALP, ALT, AST and CK-NAK observed around 120 h post-challenge for ALP, 120 h post-challenge for two animals or 168 h post-challenge for three animals for AST, 120 h post-challenge for nine animals or 168 h post-challenge for five animals for CK-NAK.

Animals 882 and 986 showed a highly significant increase of ALT at the time of euthanasia whereas all other animals demonstrated a more moderate elevation of this parameter (fold change from the baseline of ALT concentration at the time of euthanasia of 21.9 and 19.2, respectively, whereas the mean fold change of this parameter from the baseline at the time of euthanasia was 6.6 for all 12 SUDV-challenged animals).

Increase in ALT and AST may also have resulted from muscle degeneration, which is further supported by observations of moderate to marked elevation in creatinine kinase at similar timepoints.

C-reactive protein (CRP) was markedly increased for all animals, starting around 72 h post-challenge for four animals and 120 h post-challenge for eight animals until euthanasia (mean fold change from baseline at the time of euthanasia was 22.3). In particular, four animals (655, 778, 837 and 986) demonstrated an increase of CRP reaching 1.4 to 4 fold of the fold change mean for all SUDV-challenged animals. The average CRP concentration peaked by 168 h post-challenge (Figure 7).

#### 3.3.2. Hematology

The effect of SUDV infection on the blood was investigated by performing hematologic analysis on whole blood using a MS905 unit (Melet Schloesing Laboratories 3MS09007). Changes were observed in white blood cell (WBC), red blood cell (RBC), neutrophil and monocyte counts. Collectively, these observations are compatible with a severe systemic inflammatory response (Figure 8 and Appendix A).

WBC concentrations slightly increased in all animals (except for animal 778), peaking around 72 h post-challenge for seven animals, around 120 h post-challenge for two animals and around 168 h post-challenge for two animals. After this peak, WBC concentrations generally decreased until the animals reached the criteria for euthanasia.

Monocyte concentrations slightly increased in animals (except for animal 739). Monocyte counts generally increased around 72 h post-challenge (observed in 9 out of 12 animals) peaking around 72 h post-challenge for eight animals and around 168 h post-challenge for two animals, before progressively decreasing until euthanasia. One animal (986) showed a continuous increase of monocytes until the latest part of the infection.

Similarly, neutrophil counts increased and reached a peak around 72 h post-challenge for seven animals, around 120 h post-challenge for two animals and around 128 h post-challenge for one animal before decreasing. Two animals (837 and 986) showed a continuous increase of neutrophils until the latest part of infection.

A progressive decrease of RBCs, as well as significant decrease of hematocrit, hemoglobin and platelets, was detected for all animals starting from around 120 h post-challenge until the death of animals, which highlights a severe anemia and thrombocytopenia (mean fold change of all SUDV-challenged animals from mean baseline at the time of 0.8, 0.8, 0.8, and 0.4 respectively).

Moreover, a decrease in lymphocyte counts occurred from 120 h post-challenge. These concentrations stayed depressed throughout the remainder of the disease (mean fold change of all SUDV-challenged animals from mean baseline at the time of euthanasia of 0.7), except for the lymphocyte concentration values of three animals (986, 396 and 365) that rebounded before they met the criteria for euthanasia. As lymphocytes are an integral part to regulating the adaptive immune system, these findings are compatible with a state of immunosuppression.

#### 3.3.3. Coagulation

Since one of the commonly observed symptoms in filovirus infections in humans is uncontrolled bleeding from loss of coagulation, we wanted to measure the coagulation time in the cynomolgus macaques infected with SUDV. Some issues were encountered with the Istat equipment and with the quality of animals’ blood during sample collection, which resulted in an inability to generate results of the coagulation parameters at some points and for some animals.

Multiple observations for coagulation, including decreased platelet counts, slightly prolonged prothrombin time (PT) and increase of clotting times (ACT) were found and correspond with consumptive coagulopathy (Figure 9 and Appendix A).

For all SUDV-challenged animals, increased clotting time occurred late in infection or just before the animal succumbed to the disease (mean fold change of all SUDV-challenged animals from mean baseline at the time of euthanasia of 1.8). Steady decline of platelets was also observed for all animals with a significant decrease late in the infection (mean fold change of all SUDV-challenged animals from mean baseline at time of euthanasia of 0.4). Statistically significant changes consistent with coagulopathy were noted starting at 120 h post-challenge and generally increased in severity over time.

### 3.4. Viral Load

Plasmatic infectious virus was assessed by immune detection plaque assays on samples obtained on Day −7, Day −4, Day 0 (before challenge), Day 3, Day 5, Day 7, Day 10 and at the time of euthanasia. Infectious virus was not detected in any sample collected pre-challenge. The presence of infectious virus was detected on Day 3 post-challenge in one animal and on Day 5 for all twelve animals.

The highest level of infectious virus was observed on Day 5 for eight animals and Day 7 for four animals. For the three animals that first reached clinical scores for euthanasia (882, 655 and 966), the highest viral load was the final value obtained. In animals surviving longer, the viral load, as detected in the blood, decreased daily until euthanasia criteria were met (Figure 10).

#### 3.4.1. qRT-PCR Detection of Plasmatic Viral RNA

The molecular viral load, as measured by RNA copies of the region corresponding to the NP gene, were determined by RNA extraction of blood and RT-PCR on plasma samples collected on Day −7, Day −4, Day 0, Day 3, Day 5, Day 7, Day 10 and at the time of euthanasia. The lower limit of detection for the assay was 6000 copies/mL. Viral RNA was not detected in any sample collected pre-challenge. Viral RNA was first detected in plasma on Day 3 post-challenge in four animals (882, 655, 356 and 895). On Day 5 post-challenge, viral RNA was detected at quantitative concentrations in all SUDV-challenged animals. Peak values in individual animals ranged from 7.95 to 9.09 log10 copies/mL and occurred on Day 5 in eight animals and Day 7 in four animals post-challenge. In animals surviving beyond peak day of viral load, concentrations detected in the blood dropped slightly (Figure 11). When comparing the ratio of plasmatic genomic viremia to plasmatic infectious virus, approximately a two log difference (a three log difference has been observed in some cases at the early and late stages of the illness) is observed. This ratio appears to be at the lower end of the ratio generally observed [27] but is not discordant with the temporal evolution of the illness and is consistent with data from previous experiments performed with Ebolavirus on NHPs at Inserm [28].

#### 3.4.2. Next-Generation Sequencing

The effect of the host on the consensus RNA sequence during SUDV infection was investigated by purifying RNA from the Day 5 post-infection and later blood draws and creating a cDNA copy using random primers. The majority of samples had enough coverage of the reference genome (greater than 100,000 reads) to propose a consensus sequence (Table 3). All samples except those derived from two animals (356 and 396) had successful sequencing results from both Day 5 and at the time of euthanasia. One animal (882) was euthanized on Day 5, thus only one sample was sequenced for this animal. There seems to be little selective pressure on SUDV during infection as the consensus sequence for the SUDV has few changes compared to the reference genome. One variation from the reference sequence was observed at greater than 20% in samples taken at Day 5 post-challenge in two animals (656 and 655), indicating a C instead of a T at position 2609. This alternative base was a minority sequence present in all other samples where genomic sequences could be determined. The lowest amount was in the viral stock at 3.33%. This change of nucleotide at location 2609 from a T to a C results in an amino acid change of valine (V) to alanine (A) in the viral nucleoprotein at amino acid position 724 (V724A).

### 3.5. Pathology

#### 3.5.1. Necropsy and Gross Pathology

At the time of euthanasia, clinical observations showed macular skin rash for seven animals. At post-mortem, gross lesions included hemorrhages and hyperemia/congestion in multiple organs. The liver showed some discoloration, petechial hemorrhages and yellowish appearance together with generalized jaundice in some animals. The spleen showed friable consistency in most of the animals.

#### 3.5.2. Histopathology

Tissue sections stained with H&E and by in situ hybridization (RNAscope) were analyzed. Histopathological lesions and the presence of viral RNA were observed in all animals. The main histopathological lesions were observed in the spleen and lymph nodes, showing moderate to severe lymphoid depletion mostly in the follicles and hemorrhages and congestion, moderate inflammation and necrosis in the kidney, liver and lung together congestion and hemorrhages and mild presence of inflammatory infiltrates in the heart with mild congestion (Appendix A). A scoring system was used to semi-quantitatively evaluate the histopathological lesions observed in the tissue sections (Appendix A).

A semiquantitative scoring system was used to evaluate the presence of viral RNA in the RNAscope stained slides, using 0 = none; 1 = minimal; 2 = mild; 3 = moderate; and 4 = abundant (Appendix A).

### 3.6. Daily Summary of Disease

The daily summary of the disease characteristics of SUDV-challenged NHPs is described starting on the day of challenge, Day 0, through the end-of-life phase of the disease in Appendix B. An overall summary of the observed disease kinetics of infection and primary signs is found in Table 4.

## 4. Discussion

The development of medical countermeasures against rare, but potentially consequential, infectious diseases poses a unique challenge in that the traditional approach to regulatory approval, the demonstration of efficacy through well-designed clinical trials, is not feasible or ethical. An alternative regulatory path for the efficacy evaluation of medical countermeasures against these infections is the FDA Animal Rule, where the demonstration of efficacy in well-characterized and appropriate animal models can replace the clinical trials designed to demonstrate efficacy. An appropriate animal model for evaluation of medical countermeasures under the Animal Rule must meet the requirements described in FDA Guidance, “Product development under the animal rule: guidance for industry” [10], for product development using the Animal Rule. The model must be able to demonstrate that the activity of a candidate medical countermeasure in the model is predictive of similar clinical benefit if the countermeasure is used to treat the disease in humans.

The purpose of this study was to explore the feasibility of using a nonhuman-primate model where cynomolgus macaques are infected with 1000 PFU of SUDV intramuscularly as an appropriate nonclinical model for evaluation and approval of medical countermeasures against SUDV under the FDA Animal Rule. In this study, twelve animals were infected with SUDV and monitored over the course of the infection for changes in clinical signs, body weight, temperature, viremia, hematology, clinical chemistry and coagulation. Although other natural history studies supported by BARDA have included a mock-infected cohort to establish baseline values for biomarkers during the infection, the Institutional Animal Care and Use Committee (IACUC) determined that data collected prior to challenge was sufficient to establish biomarker baselines and the inclusion of cynomolgus macaques for only blood collection was not ethically justified. The maximum number of animals that can be housed at one time in the Inserm BSL4 laboratory space is twelve.

This model was chosen based on results from previous studies that employed a 1000 PFU intramuscular challenge in cynomolgus macaques as the primary model to evaluate candidate vaccines against Ebola (EBOV) [29]. The ERVEBO vaccine [30], which demonstrated clinical benefit in a ring vaccination strategy during the outbreaks in West Africa, was initially tested and was consistently shown to be effective in the 1000 PFU challenge model in cynomolgus macaques [12,31]. We hypothesized that an analogous model using 1000 PFU of SUDV as the challenge agent would also be appropriate to evaluate the efficacy of candidate vaccines against SUDV. Recently, a SUDV IM challenge natural history study compared the disease course in both rhesus and cynomolgus macaques [21]. However, this study did not report telemetry or body weight changes during the course of SUDV infection. Overall, the results from this natural history study, including time until death and clinical pathology, are similar.

The course of the disease after infection included a refractory period with few signs of disease, followed by increasing clinical scores starting at Day 3 post-challenge for some animals and by Day 5 in all twelve animals. All twelve animals reached a clinical score (15 or higher) that resulted in euthanasia between five and ten days post-challenge. The median time to death was 182 h post-challenge. The primary clinical signs of disease included sustained fever, reduced physical activity, macular skin rash, and dehydration. The combined effect of multiple clinical signs ultimately coincided with the rapid clinical decline of the animals. Similar clinical signs have been observed in human SUDV cases.

One of the traits of filovirus infection is a sustained fever. To monitor temperature, two systems were employed, the DSI PhsyioTel Digital Telemetry and the StarOddi DST micro-T transmitters. Fever, defined as body temperature > 1.7 °C above baseline for longer than two hours, was detected using the DSI telemetry system by Day 3 post-challenge in eleven of the twelve animals, with the remaining animal developing fever early on Day 4. Analysis of the StarOddi body temperature data revealed an onset of fever by Day 3 for all twelve animals. Sustained fever was noted in all SUDV-challenged animals with hyperpyrexia observed from Days 3–6 post-challenge. Additionally, a loss of thermal regulation and rapid drop in temperature was observed in 9 of 11 animals with the DSI implants, and 9 of 12 animals with the StarOddi telemetry system at time of euthanasia. The loss of thermal regulation may provide another symptom in this animal model for discriminating between those animals that are responding to a medical countermeasure treatment and those where euthanasia to reduce animal suffering is appropriate. The results from this study indicate that either implant can serve to provide temperature information and will be useful tools in the application of this model in efficacy studies. However, the DSI telemetry system, but not StarOddi transmitters, provides real time telemetry.

The blood draws were used to measure biomarkers associated with clinical chemistry, hematology and coagulation. In all animals, evidence of severe disease was observed at later times post-challengewith enzymes indicative of systemic inflammation, coagulopathy, lymphopenia, renal tubular necrosis, hepatocellular degeneration and necrosis increasing at 120 h after infection or later. The kinetics of the appearance of necrotic enzymes and coagulation abnormalities were similar to the observations in the EBOV nonhuman primate animal models [31]. Similar symptoms were also detected in SUDV infections in humans, suggesting that infection in nonhuman primates resembles human infection. This supports the use of the nonhuman primate animals as a model that is utilized to predict the clinical efficacy of medical countermeasures.

Clinical pathology assessments were conducted from baseline blood samples obtained from animals prior to challenge. Comparison of these baseline values with values determined at different timepoints during the experiment reveal manifestations of disease biomarkers with statistically significant changes attributable to SUDV infection. A significant increase in CRP, a marker of tissue damage, was observed in all SUDV-challenged animals, detectable on Day 3 (*n* = 4 animals) or Day 5 (*n* = 8 animals) post-challenge. Similarly, increase of liver enzymes was observed in all animals beginning Day 4 post-challenge. Creatinine and urea concentration increased five days and seven days post-challenge, respectively, peaking at the latest time of infection for all animals.

Additional uniform clinical pathology changes observed included a progressive, continuous decrease of red blood cells as well as a significant decrease in lymphocyte count, hematocrit, hemoglobin and platelets starting at Day 4 post-challenge. Lymphocyte count increases were observed from Day 9 in three of the five remaining animals.

A slight increase in the concentrations of WBC, monocytes, and neutrophil was observed starting from Day 3 post-infection before decreasing throughout the remainder of the study (observed for *n* = 11 animals, *n* = 11 animals and *n* = 10 animals, respectively). Late in the infection, increasing concentrations of these three parameters was detected in three animals that were euthanized at 237.93 h, 238.16 h, and 239 h post-challenge. A decrease in the neutrophil concentration was also detected at the latest time point of the infection for two other animals that were euthanized at 16.08 h and 215.08 h post-challenge.

At the latest timepoint in infection, seventeen clinical pathology parameters showed statistically significant changes from baseline in all SUDV-challenged animals. Comparison of the clinical pathology endpoint of males and females revealed statistically significant changes of fifteen of these parameters for males and twelve for females (Table 5).

SUDV viral load was measured using both biological and molecular approaches. Plasma from each blood draw was incubated on Vero E6 cells and processed by immunostaining of SUDV particles. One animal had detectable infectious blood-borne virus on Day 3 post-challenge while all animals tested positive for circulating infectious virus at the next blood draw on Day 5 post-challenge. The peak level of infectious circulating virus was observed on Day 5 or Day 7 post-challenge with levels quickly dropping for those animals that survived beyond Day 7. It is likely that the host immune response is active by Day 7 post-challenge. Using a molecular approach, the presence of RNA copies of the NP gene was measured by qRT-PCR on RNA extracted from blood draws. The first evidence for SUDV RNA was detected on Day 3 for four of twelve animals with all animals testing positive for SUDV RNA at Day 5. The levels of RNA copies of the NP gene mirrored the levels of infectious virus in that the maximum values were seen at Day 5 or Day 7 post infection and dropped after Day 7 until euthanasia criteria were met.

Filoviruses are known to undergo selective pressure when passed in cell lines [32]. The RNA from blood after Day 5 post-challenge and from the terminal bleed at the time of euthanasia was used to construct a cDNA library using random primers and used as a substrate for extensive DNA sequencing. The results showed that the reference sequence was the predominant sequence at all times during the infection. A nucleotide change of T to C at position 2609 was observed in the viral stock and in samples from eleven out of twelve animals. It remains unclear if this sequence change is relevant as in some cases the relative frequency increases with later analysis and in some cases the frequency decreases. This sequence, and the resulting conversion of a valine to an alanine at position 724 in the NP gene, was also observed in SUDV infections executed as part of other natural history studies (data not shown). The approach of sequence analysis used in this study is an effective means of determining the consensus sequence but is not as accurate in describing the RNA molecules late in infection. Analysis of the RNA genomes during the course of Ebola infection has shown that defective truncated panhandle genomes accumulate late in infection [33]. In another SUDV natural history study performed at another BSL-4 facility, viral RNA was enriched by selecting out ribosomal and messenger RNA and analyzing the resulting cDNA library for evidence of defective genomes [34]. The predominant forms of the remaining RNA molecules were defective panhandle genomes (manuscript in preparation). The formation and persistence of defective genomes has been suggested as an explanation for persistence [35].

## 5. Conclusions

The results from this study provide a detailed characterization of the time course of the natural history of disease in cynomolgus macaques challenged with 1000 PFU SUDV via IM injection. This challenge dose was uniformly lethal with clear and easily observable signs of clinical disease and pathology such as viremia, fever, lymphoid depletion and systemic inflammation. Additionally, several aspects of the disease observed in this model were consistent with those observed in human cases of SUDV infection. In all, these data support the use of cynomolgus macaques as a model for testing and evaluation of medical countermeasures against disease caused by SUDV.

## Figures and Tables

**Figure 1 vaccines-10-00963-f001:**
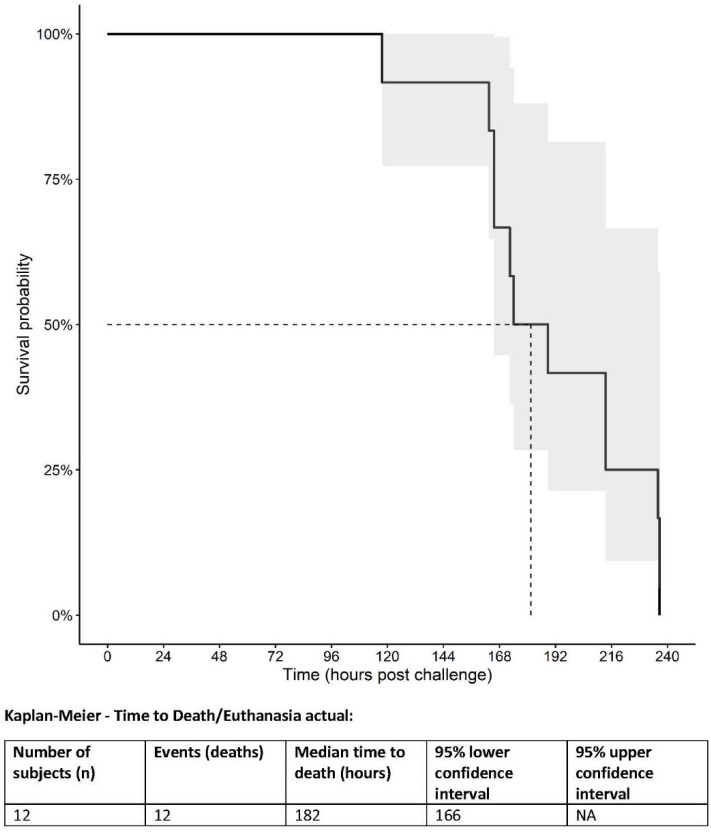
Kaplan–Meier plot of survival of SUDV-challenged cynomolgus macaques. Quantile estimate: Median time (IQR): 188.83 h (165.67 to 236.00).

**Figure 2 vaccines-10-00963-f002:**
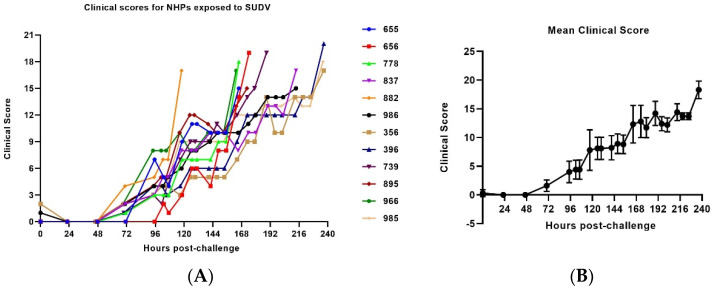
Clinical disease scores in SUDV-challenged NHPs. (**A**) Clinical disease score evolution of individual SUDV challenged animals throughout the infection. (**B**) Average clinical score evolution throughout the infection. (Day 0–Day 5 a.m.: *n* = 12 animals, Day 5 p.m.–Day 7 a.m.: *n* = 11 animals, Day 7 p.m.: *n* = 8 animals, Day 7 night–Day 8 a.m.: *n* = 6 animals, Day 8 a.m.–Day 9 a.m.: *n* = 5 animals, Day 9 p.m.–Day 10: *n* = 3 animals). The error bars represent the standard deviation (SD).

**Figure 3 vaccines-10-00963-f003:**
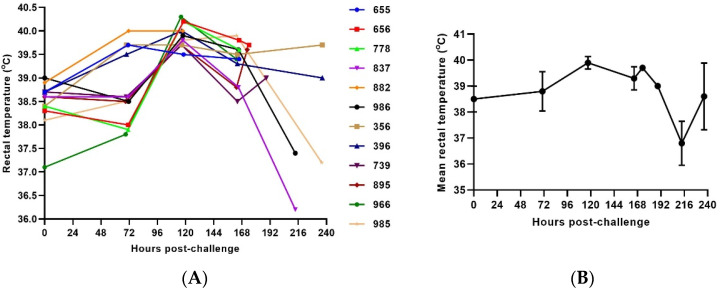
Rectal Temperature in SUDV challenged NHPs. (**A**) Rectal temperature in individual SUDV challenged animals versus time. (**B**) Group mean rectal temperature. The error bars represent the standard deviation (SD).

**Figure 4 vaccines-10-00963-f004:**
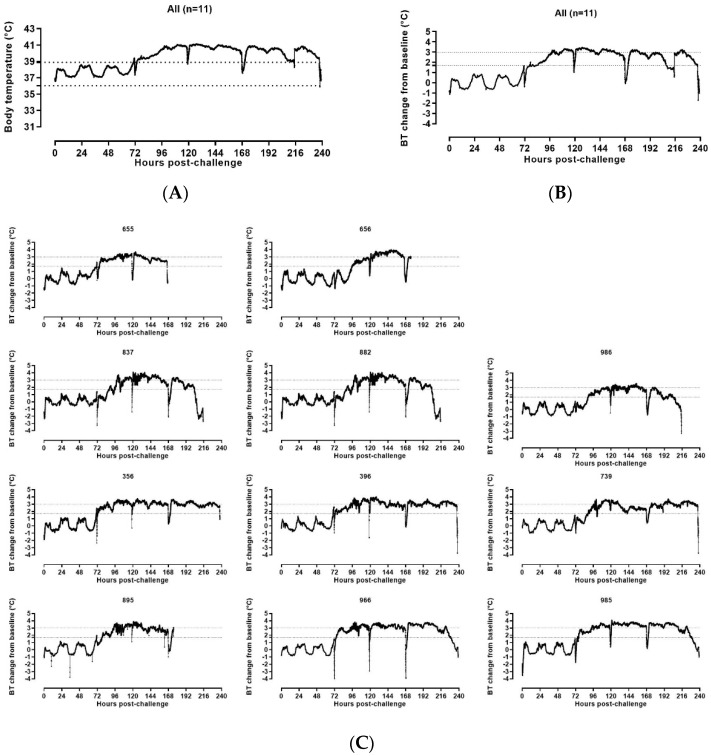
DSI telemetry body temperature changes in SUDV-challenged NHPs. (**A**) Group means DSI telemetry-based Body Temperature (BT) versus time. The dashed lines indicate the baseline minimum and maximum BT values. (**B**) Group mean DSI telemetry-based BT change from baseline versus time. The dashed lines indicate the BT change from baseline corresponding to fever (≥1.7 °C) and hyperpyrexia (≥3 °C). (**C**) DSI telemetry-based body temperature (BT) change from baseline in individual SUDV-challenged animals versus time. The straight dashed lines indicate the BT change from baseline corresponding to fever (≥1.7 °C) and hyperpyrexia (≥3 °C).

**Figure 5 vaccines-10-00963-f005:**
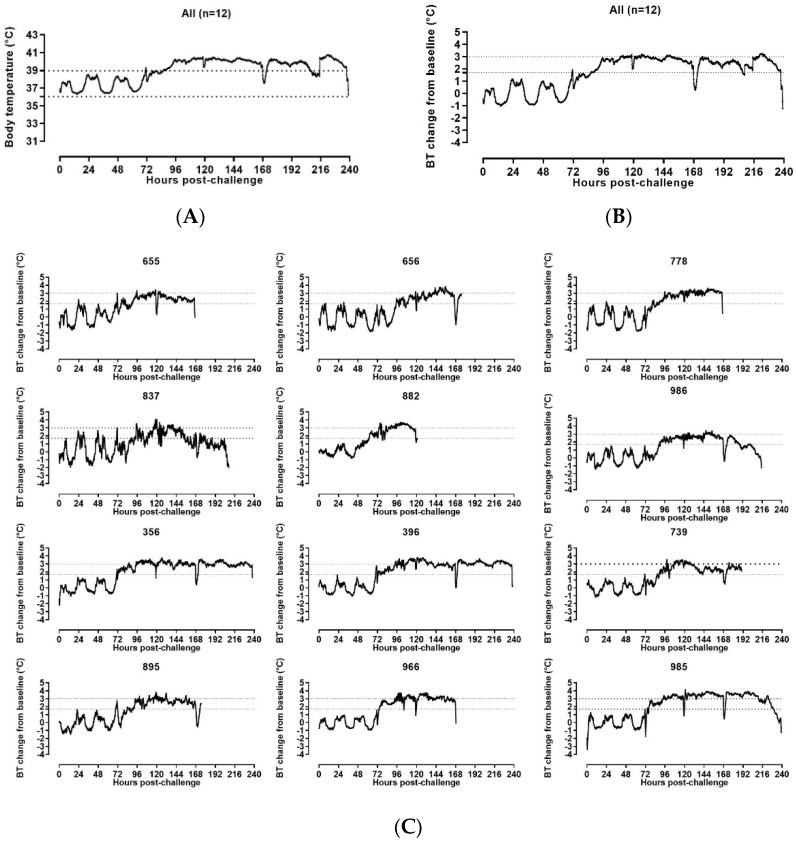
Star Oddi-based body temperature changes in SUDV-challenged animals. (**A**) Group mean StarOddi telemetry-based Body Temperature (BT) versus time. The straight dashed lines indicate the baseline minimum and maximum BT values. (**B**) Group mean StarOddi telemetry-based BT change from baseline versus time. (**C**) Star Oddi telemetry-based BT change from baseline in individual SUDV challenged animals versus time. (**B**,**C**): The straight dashed lines indicate the BT change from baseline corresponding to fever (≥1.7 °C) and hyperpyrexia (≥3 °C)).

**Figure 6 vaccines-10-00963-f006:**
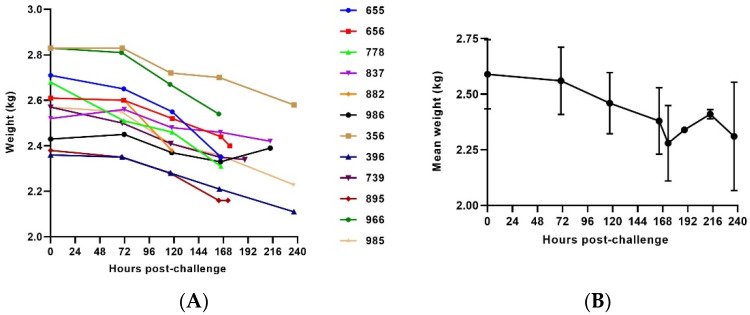
Body weight in SUDV-challenged NHPs. (**A**) Body weight evolution post-challenge. (**B**) Average weight. The error bars represent the standard deviation (SD).

**Figure 7 vaccines-10-00963-f007:**
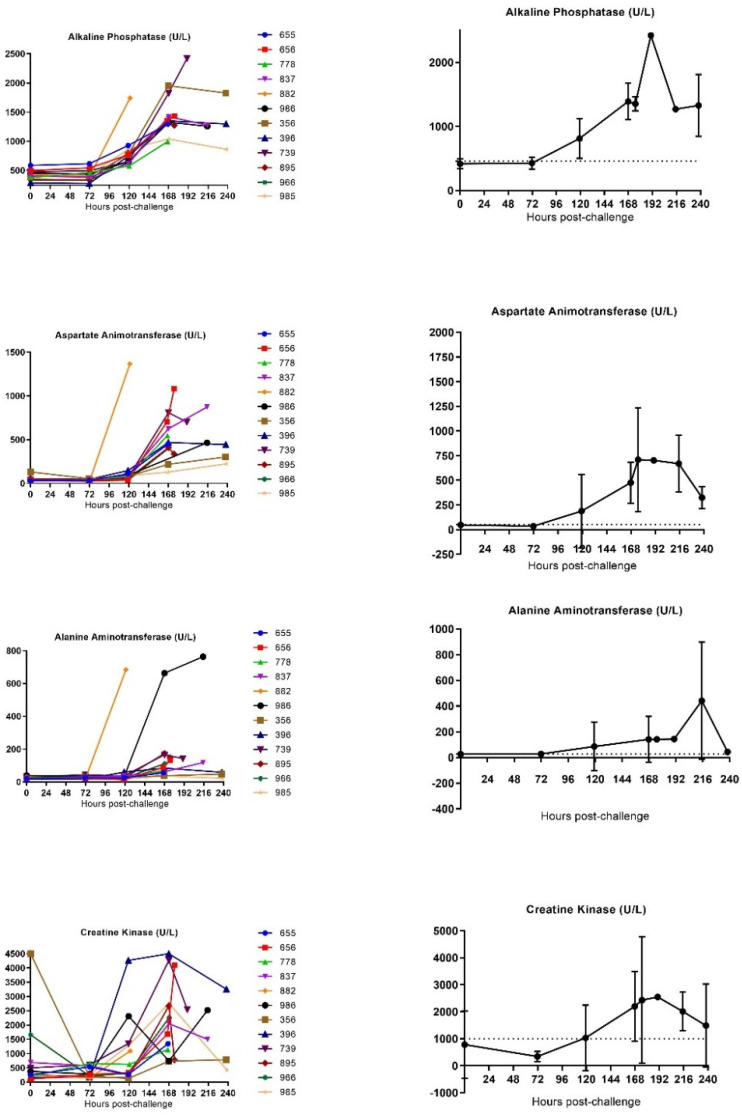
Biochemical analysis in SUDV-challenged animals. (**A**) Individual biochemical parameter evolution versus time. (**B**) Group mean biochemical parameter evolution versus time. Average measurements taken before exposure serve as baseline (represented by horizontal straight dashes lines). Error bars represent the standard deviation (SD) of the mean.

**Figure 8 vaccines-10-00963-f008:**
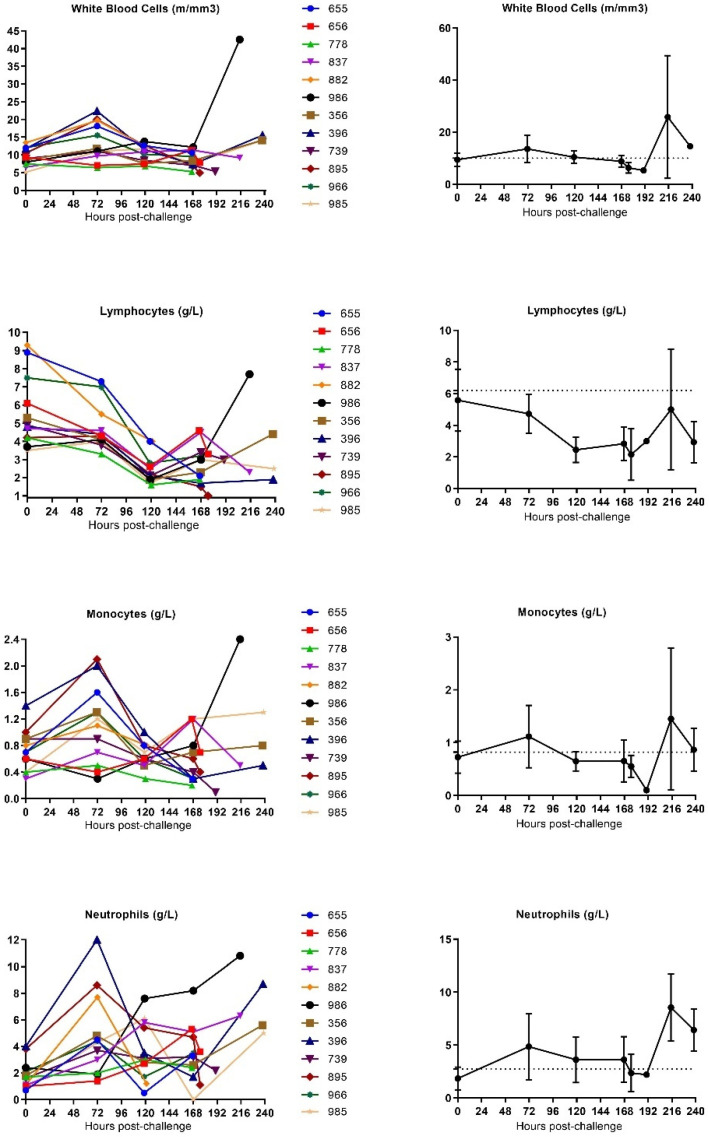
Hematological analysis in SUDV-challenged animals. (**A**) Individual hematological parameter evolution versus time. (**B**) Group mean hematological parameter evolution versus time. Average measurements taken before exposure serve as baseline (represented by horizontal straight dashes lines). Error bars represent the standard error of the mean.

**Figure 9 vaccines-10-00963-f009:**
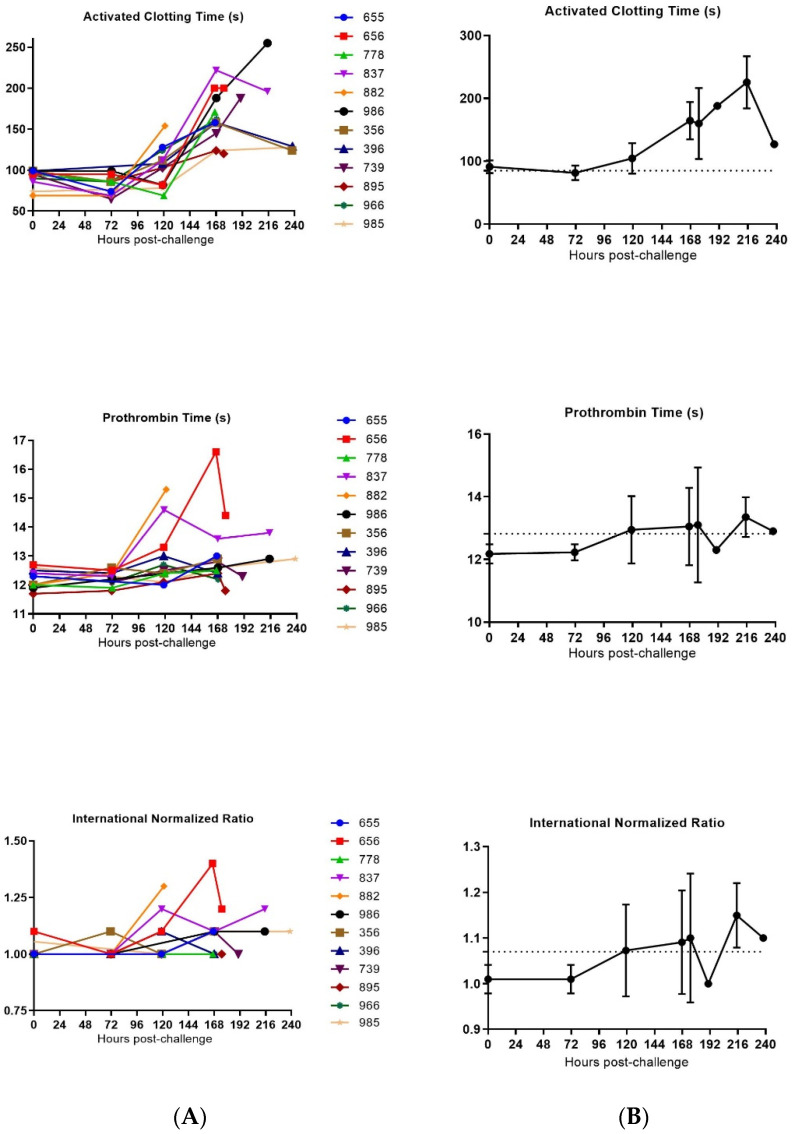
Coagulation parameters in SUDV-challenged animals. (**A**) Individual coagulation parameters evolution versus time. (**B**) Group means coagulation parameters versus time. Average measurements taken before exposure serve as baseline (represented by horizontal straight dashes lines). Error bars represent the standard error of the mean.

**Figure 10 vaccines-10-00963-f010:**
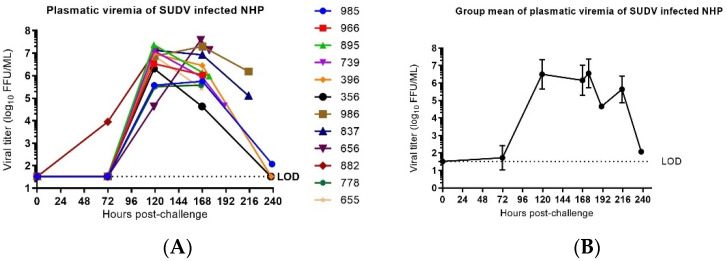
Plasmatic infectious virus in SUDV-challenged NHP. (**A**) Plasmatic viral infectious load in individual SUDV-challenged animals versus time. Viral load represents the mean of two titration assay replicates and are expressed in logarithm decimal FFU/mL. Time course is expressed in hour(s) post-challenge (pc). LOD is the limit of detection of the immunological detection plaque titration assay (=1, 52 log_10_ FFU/mL). (**B**) Group mean ± SD of SUDV infectious viral load versus time. Time course is expressed in days post-challenge (pc).

**Figure 11 vaccines-10-00963-f011:**
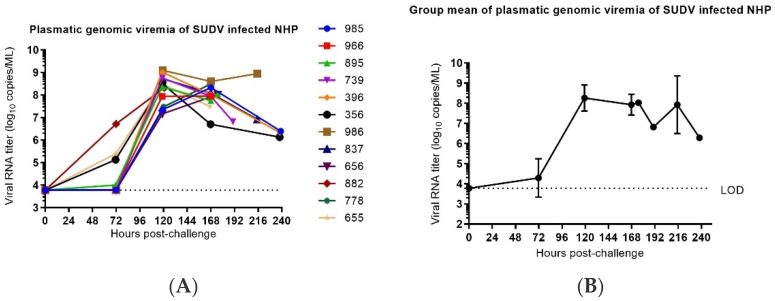
Plasmatic genomic viremia in SUDV-challenged NHPs. (**A**) Plasmatic viral RNA concentration in individual SUDV-challenged animals versus time. vRNA load represents the mean to two titration assay replicates and are expressed in logarithm decimal copies/mL. Time course is expressed in hours post-challenge (pc). LOD is the limit of detection of the PCR assay (=3.78 log_10_ copies/mL). (**B**) Group mean ± SD of plasmatic viral RNA versus time. Time course is expressed in days post-challenge (pc).

**Table 1 vaccines-10-00963-t001:** Clinical Scoring.

Parameter	Description	Score Points and Number of Levels ^1^
Temperature	Delta temperature reference	Four levels: from 0 to over than a 2.2 °C increase (0 to 3 score points)
Weight	Delta weight reference	Four levels: from ΔW ≤ 5.4% to ΔW < 10%weight loss (from 0 to 5 score points)
Dehydration	General dehydration	Two levels: from NTR to dehydration (0 to 1 score point)
Bleeding	Presence of hemorrhage	Two levels: from NTR to bleeding observations (0 to 3 score points)
Petechia	% of body	Four levels: from NTR to observation of petechia on more than 50% of body (0 to 3 score points)
Stool	Stool appearance	Four levels: from NTR mucous and fibrinous diarrhea (0 to 5 score points)
Responsiveness	General behavior of the animal	Six levels: from normal activity to moribund (0 to 15 score points)

^1^ (NTR, nothing to report).

**Table 2 vaccines-10-00963-t002:** Survival time of the SUDV-challenged cynomolgus macaques.

Animal ID	Time Post-Infection until Death (Days)	Survival Time (Hours)
655	7	165.7
656	7	174.2
778	7	165.7
837	9	213.5
882	5	117.8
986	9	213.5
356	10	236.7
396	10	236.7
739	8	188.8
895	7	172.5
966	7	163.5
985	10	236.0

**Table 3 vaccines-10-00963-t003:** Percentage of reads containing alternative base at location 2609.

Animal ID	Sample Collection (Day Post-Infection)	NGS Code Assigned	Reads for Ref T (%)	Reads for Alt. C (%)
N/A Viral stock NR50733	N/A	13.S	96.55	3.44
882	Day 5 T	8.A	78.33	21.46
655	Day 5	3.5	78.29	21.7
	Day 7 T	3.A	89.3	10.49
778	Day 5	6.5	85.38	14.53
	Day 7 T	6.A	91.99	8
966	Day 5	10.5	84.75	15.25
	Day 7 T	10.A	93.41	6.4
656	Day 5	4.5	77.7	22.29
	Day 7 p.m. T	4.A	82.79	17.14
895	Day 5	9.5	89.53	10.32
	Day 7 p.m. T	9.A	81.59	18.4
739	Day 5	5.5	82.86	16.98
	Day 8 T	5.A	92.91	7.017
837	Day 5	7.5	90.52	9.38
	Day 9 T	7.A	90.17	9.83
986	Day 5	12.5	93.16	6.84
	Day 9 T	12.A	91.61	8.38
356	Day 5	1.5	-	-
	Day 10 T	1.A	99.88	0
396	Day 5	2.5	95.1	4.82
	Day 10 T	2.A	-	-
985	Day 5	11.5	82.5	17.5
	Day 10 T	11.A	87.27	12.72

**Table 4 vaccines-10-00963-t004:** Summary of observed kinetics of infection and of key disease characteristics in SUDV-challenged animals.

Manifestations	Disease Kinetics	Frequency
Incubation Period ^a^	DSI temperature data-based (*n* = 11): 2.577–4 days (61.85–96.04 h PC) StarOddi temperature data-based (*n* = 12): 2.576–3.965 days (61.83–95.17 h PC)	
Symptom onset (fever) to Death	DSI temperature data-based (*n* = 11): 2.46–6.89 days (59.02–165.40 h) StarOddi temperature data-based (*n* = 12): 2.46–6.89 days (59.04–65.34 h)	
Mortality	100% within 5–10 days PC	*n* = 12
Median survival Time (IQR)	7.92 days PC (190.42 h PI)	
Plasma viral RNA		*n* = 12
First detectable	Coincident with symptoms onset (starting D3 PC (71.43−71.88 h PC) for *n* = 4 animals At D5 PC (118.69–121.15 h PC) detectable for *n* = 12	
Plasma viremia		*n* = 12
First detectable	Coincident with symptoms onset Starting D3 PC (71.93 h PC) for *n* = 1 animal At D5 PC(118.69–121.15 h PC) detectable for *n* = 12 animals	
Peak	At D5 PC (*n* = 8) 0.95–2.46 days after symptoms onset (DSI: 23.04–59.3 h after fever onset/ Star Oddi: 23.75–59.32 h after fever onset) At D7 PC (*n* = 4) 2.9–4 days after symptoms onset (DSI: 70.83–95.75 h after fever onset/ Star Oddi: 71.33–98.50 h after fever onset) Viral RNA: 7.95 to 9.09 log10 copies/mL Infectious virus: 5.19 × 10^5^ to 3.27 × 10^7^ ffu/mL	
Clinical Disease Signs		
	Fever	*n* = 12
	Diarrhea (observed from D3 (69:00–70:35 h) PC);	*n* = 10
	Dehydration (detected from D5 (116:10–118:20 h) PC for *n* = 10/at D8 (188:50 h) PC for *n* = 1)	*n* = 11
	Rash (Observed from D5 (116:10–118:20 h) PI for *n* = 1 and from D7 (163:30–165:40 h) PC for *n* = 7)	*n* = 8
	Change in responsiveness score starting from D4 (94–102:20 h) PC (*n* = 4) and D5 (116:10–126:05 h) PC (*n* = 8)	*n* = 12
	Prostration (responsiveness score of 10) observed at D7 (172:30–174:10 h) PC for *n* = 2, D8 (188:50–189:45 h) PC for *n* = 3, D9 (213:30 h) PC for *n* = 1 and D10 (236:00 h) PC for *n* = 1	*n* = 7
	Mucous and fibrinous diarrhea (D10 PC)	*n* = 1
Clinical Pathology		
	Systemic inflammation Liver dysfunction Renal dysfunction Coagulopathy Lymphocytolysis Anemia and thrombocytopenia	
Biochemical parameters	CRP: Increase from D3 (71.43–72 h) PI for *n* = 4 and from D5 (118.68–121.15 h) PI for *n* = 8	*n* = 12
	Creatinine: Increase from D5 (118.68–121.15 h) PI	*n* = 12
	UREA: Increase from D7 (166.5–168.5 h) PI	*n* = 12
	ALP: Increase from D5 (118.68–121.15 h) PC	*n* = 12
	ALT: Increase from D5 (118.68–121.15 h) PC for *n* = 2 and from D7 (166.5–168.5 h) PC for *n* = 8	*n* = 10
	AST: increase from D5 for *n* = 6 and from D7 (166.5–168.5 h) PI for *n* = 7	*n* = 11
Hematological parameters	RBC, hematocrit, hemoglobulin, platelet: Decrease from D5 (118.68–121.15 h) PC	*n* = 12
	Lymphocytes: decrease from D5 (118.68–121.15 h) PC	*n* = 12
Coagulation parameters	ACT: Increase from D5 (118.68–121.15 h) PC for *n* = 8, from D7(166.5–168.5 h) PC for *n* = 4	*n* = 12
	PT: increase from D5 (118.68–121.15 h) PC for *n* = 8, from D7 (166.5–168.5 h) PC for *n* = 2	*n* = 10/10 *

^a^ The incubation period is defined as the time from virus exposure to fever onset. * instrumental difficulties precluded baseline values on PT, for 2 out of 12 SUDV-exposed animals. Time course is expressed in D (days) or h (hours) PC (post-challenge).

**Table 5 vaccines-10-00963-t005:** Statistical analyses of the clinical pathology changes from the baseline values in terminal samples of SUDV-challenged animals. To determine statistically significant changes at the time of euthanasia the mean of terminal values were compared to the mean of baseline-value using the Wilcoxon matched-pairs signed ranks tests. Mean values of the complete group of animals (*n* = 12), of the group of male animals (*n* = 6) and of the group of female animals (*n* = 6) have been considered. Differences from baseline were considered to be statistically significant at *p*-values < 0.05.

	**Biochemical Analyses**															
	**Urea**		**Creatinine**		**ALP**		**ALT**		**AST**		**CK-NAK**		**CRP**			
**Wilcoxon matched-pairs signed rank test**		*p* value		*p* value		*p* value		*p* value		*p* value		*p* value		*p* value		
Mean values (*n* = 12) changes from the mean baseline	***	0.0005	***	0.0005	***	0.0005	***	0.001	***	0.0005	*	0.0269	***	0.0005		
Male mean values changes from the baseline	*	0.0313	*	0.0313	*	0.0313	*	0.0313	*	0.0313	*	0.0313	*	0.0313		
Female mean values changes from the baseline	*	0.0313	*	0.0313	*	0.0313	ns		*	0.0313	ns		*	0.0313		
	**Hematological Analyses**															
	**WBC**		**Monocytes**		**Neutrophils**		**RBC**		**Hematocrit**		**Haemaglobulin**		**Platelets**		**Lymphocytes**	
**Wilcoxon matched-pairs signed rank test**		*p* value		*p* value		*p* value		*p* value		*p* value		*p* value		*p* value		*p* value
Mean values (*n* = 12) changes from the mean baseline	ns		ns		**	0.0093	***	0.0005	***	0.0005	***	0.0005	***	0.0005	*	0.0161
Male mean values changes from the baseline	ns		ns		*	0.0313	*	0.0313	*	0.0313	*	0.0313	*	0.0313	ns	
Female mean values changes from the baseline	ns		ns		ns		*	0.0313	*	0.0313	*	0.0313	*	0.0313	*	0.0313
	**Coagulation Analyses**															
	**ACT**		**PT**		**INR**		**Platelets**									
**Wilcoxon matched-pairs signed rank test**		*p* value		*p* value		*p* value		*p* value								
Mean values (*n* = 12) changes from the mean baseline	***	0.0005	*	0.0156	*	0.0313	***	0.0005								
Male mean values changes from the baseline	*	0.0313	*	0.0313	ns		*	0.0313								
Female mean values changes from the baseline	*	0.0313	ns		ns		*	0.0313								

Statistically significant changes amongst the entire group of SUDV-exposed animals (*n* = 12). Statistically significant changes amongst the entire group of males (*n* = 6). Statistically significant changes amongst the entire group of females (*n* = 6). No statistically significant changes (ns). Significance level set to *p*-values: <0.05 (*), <0.01 (**), <0.001 (***), <0.0001 (****).

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
