# Peer review of "Natural History of Sudan ebolavirus to Support Medical Countermeasure Development"

_vaccines, 2022, doi:10.3390/vaccines10060963_

Round 1

Reviewer 1 Report

The manuscript by Carbonelle et al. “Natural History of Sudan ebolavirus to Support Medical Countermeasure Development” contains a detailed description of the virological, clinical and pathological findings in a macaque model of SUDV infection. The work is well written and presented in a clear and straightforward manner, the whole work is quite informative.

Specific comments:

1: There are indeed previous experience with SUDV infection in NHP. This should be adequately referenced and discussed. The number of animals (12) was high for this type of work under BSL4 conditions but there was a single agent and a single dose experiment, so the contribution of the present work as compared with the previous studies should be explained.

2: Related to the infection inoculum it would be important to know the passage and/or the GP 7 vs 8-uridine genotype of the stock.

3: In figures 10 and 11 it is shown the temporal evolution of blood viral load of SUDV both by cell culture and PCR. From the graphs, the ratio of RNA copies per PFU seems rather low (@10) when in general it has been shown for most viruses including Filo that the ratio of non-infectious vs infectious particles is at least in a range of 3-4 orders of magnitude, this should be also commented

Author Response

We thank the reviewer for your comments which were helpful in revising our paper. We have included our responses in blue below and a revised copy of the paper is attached.

1.) There are indeed previous experience with SUDV infection in NHP. This should be adequately referenced and discussed. The number of animals (12) was high for this type of work under BSL4 conditions but there was a single agent and a single dose experiment, so the contribution of the present work as compared with the previous studies should be explained.

We thank the reviewer for your helpful comments. We have revised the introduction of the paper to include references to previous published SUDV challenge studies in nonhuman primates and outlined the additional contribution of our paper as compared to previous studies. Additionally, we have included a short discussion of a recently published SUDV natural history study (published June 3, 2022) in our discussion section.

2.) Related to the infection inoculum it would be important to know the passage and/or the GP 7 vs 8-uridine genotype of the stock.

The challenge stock was a third cell culture passage (P3) and had a 100% 7-uracil (U) genotype at the glycoprotein editing site start at nucleotide 6925.  We have revised the materials and methods section, under “2.7 Challenge Material” to include this information.

3.) In figures 10 and 11 it is shown the temporal evolution of blood viral load of SUDV both by cell culture and PCR. From the graphs, the ratio of RNA copies per PFU seems rather low (@10) when in general it has been shown for most viruses including Filo that the ratio of non-infectious vs infectious particles is at least in a range of 3-4 orders of magnitude, this should be also commented.

When comparing the ratio of plasmatic genomic viremia to plasmatic infectious virus in our paper, approximately a two log difference (a three log difference has been observed in some cases of the early and late stage of the illness) is observed. This ratio appears to be at the lower end of the ratio generally observed  but is not discordant with the temporal evolution of the illness and consistent with data from previous experiments performed with Ebolavirus on NHPs at Inserm.  This information and several references have been added to the paper in section “3.4 viral load”.

Reviewer 2 Report

The article “Natural History of Sudan ebolavirus to Support Medical Countermeasure Development” details the results of an experimental infection study where 12 cynomolgus macaques were inoculated with Sudan ebolavirus. This manuscript is very well written and describes in great detail the course of the resulting infections in the macaques. Not only were the authors were very thorough in their description of the various tropisms brought on by infection with Sudan ebolavirus, their use of telemetric surveillance to acquire activity and body temperature data provided real-time effects of infection that will no doubt be very useful in vaccines studies. The authors clearly demonstrate through this study that cynomolgus macaques are an appropriate animal model to study the effects of Sudan ebolavirus infection as well as an appropriate model for the study of medical countermeasures. My only suggestion would be the addition of a table and short description of viral loads described as TCID50 equivalents/mg of visceral tissue if they are available. This could be done as a supplement or in the main text. I feel this would benefit future research involving medical countermeasure against Sudan ebolavirus. This is not a requirement however and I will defer to the discretion of the editors and authors.

Author Response

We thank the reviewer for your comments which were helpful in revising our paper. Our response to your question is below in blue.

  • The article “Natural History of Sudan ebolavirus to Support Medical Countermeasure Development” details the results of an experimental infection study where 12 cynomolgus macaques were inoculated with Sudan ebolavirus. This manuscript is very well written and describes in great detail the course of the resulting infections in the macaques. Not only were the authors were very thorough in their description of the various tropisms brought on by infection with Sudan ebolavirus, their use of telemetric surveillance to acquire activity and body temperature data provided real-time effects of infection that will no doubt be very useful in vaccines studies. The authors clearly demonstrate through this study that cynomolgus macaques are an appropriate animal model to study the effects of Sudan ebolavirus infection as well as an appropriate model for the study of medical countermeasures. My only suggestion would be the addition of a table and short description of viral loads described as TCID50equivalents/mg of visceral tissue if they are available. This could be done as a supplement or in the main text. I feel this would benefit future research involving medical countermeasure against Sudan ebolavirus. This is not a requirement however and I will defer to the discretion of the editors and authors.

We thank the reviewer for the comments.  We are unable to provide a table of viral loads of visceral tissue as tissues were not sampled for infectious viral load in this study.